∂ | **Open Peer Review** | Mycobacteriology | Research Article

# Assembly of the *Mycobacterium tuberculosis* type VII ESX-1 secretion system in *Mycobacterium smegmatis* identifies a new transcriptional activator of *esx-1* genes and a novel TB vaccine

Slim Zriba,[1,2] Ze Long Lim,[1] Marlene Snider,[1] Nirajan Niroula,[1] Marie Hardouin,[1] Jeffrey M. Chen[1,2]

**ABSTRACT** *Mycobacterium tuberculosis* (*M. tb*) uses its type VII secretion system (T7SS) ESX-1 to export immunogenic, virulence-mediating protein effectors. In this study, the fast-growing, non-pathogenic model mycobacteria *Mycobacterium smegmatis* mc²-155 was engineered to express the *M. tb* T7SS ESX-1 system. We found that *M. smegmatis* transformed with *M. tb esx-1* locus genes only, as well as *M. smegmatis* transformed with *M. tb esx-1* and *espACD* operon genes (designated MSX-1), produces and secretes the *M. tb* ESX-1 protein effectors EsxA, EsxB, and EspB. However, the abundance of these proteins was higher inside the cell and culture filtrate of the MSX-1 strain. Although ESX-1 is critical for *M. tb* pathogenesis, expression of *M. tb* ESX-1 did not make the recombinant *M. smegmatis* strains virulent in macrophages. Serendipitously, transformation of *M. smegmatis* with a modified *esx-1* locus in this study revealed *rv3860*, a gene of previously unknown function, to be required for the transcription of *pe35*, *ppe68*, *esxB*, and *esxA* genes. Finally, mice vaccinated with MSX-1 were found to be as protected as mice vaccinated with *Mycobacterium bovis* BCG against *M. tb* infection, without becoming sensitized to tuberculin. These results show that a functional *M. tb* ESX-1 system can be assembled in *M. smegmatis* to uncover novel facets of the secretion machinery and that the modified *M. smegmatis* strain can function as a tuberculosis (TB) vaccine. Unlike BCG, however, its deployment may be compatible with tests currently used to diagnose TB.

**IMPORTANCE** In this study, we modified *Mycobacterium smegmatis*, which is often used as a surrogate model organism in mycobacterial research, to produce and assemble a functional *Mycobacterium tuberculosis* (*M. tb*) ESX-1 protein secretion system. One such *M. smegmatis* strain named MSX-1 was found to make a functional *M. tb* ESX-1 system without becoming virulent. And in using *M. smegmatis* as a chassis to study the ESX-1 system, we found that *rv3860*, an *M. tb* gene of previously unknown function, is needed for the production of key ESX-1 proteins. Finally, mice vaccinated with MSX-1 were as protected from tuberculosis (TB) as mice given BCG, the only approved TB vaccine. Notably, we found that unlike BCG, MSX-1 does not sensitize mice to the antigens used in existing TB diagnostic tests. These observations, taken together, highlight the utility of *M. smegmatis* as a chassis to study the *M. tb* ESX-1 secretion machinery.

**KEYWORDS** *M. tuberculosis*, *M. smegmatis*, type-7 secretion system, ESX-1

Mycobacterium tuberculosis (*M. tb*) and other tuberculosis (TB)-causing members of the *M. tb* complex (MTBC) export virulence effector proteins through five functionally distinct forms of the type VII protein secretion system—ESX-1 to ESX-5 (1). Of these, ESX-1 is the most critical for MTBC virulence and immunopathogenesis, and based on structural studies, it is predicted to consist of a multimeric

**Peer Reviewer** Giovanni Delogu, Catholic University of the Sacred Heart, Rome, Rome, Italy

Address correspondence to Jeffrey M. Chen, jeffrey.chen@usask.ca.

Slim Zriba and Jeffrey M. Chen are named as co-inventors of MSX-1 as a TB vaccine in an International Patent application.

See the funding table on p. 16.

membrane-spanning translocon core made up of the $EccB_1$, $EccCa_1$, $EccCb_1$, $EccD_1$, and $EccE_1$ proteins (2, 3). This translocon core works with other ESX-1-associated proteins in an energy-dependent and orchestrated manner to export EspA, EspB, EspC, EsxA, and EsxB virulence effectors (1, 3–12), which are also highly immunodominant protein antigens (1, 4, 11, 13–15). While EsxA, EsxB, EspA, and EspC exhibit mutually co-dependent secretion (5, 6, 8, 10), the secretion of EspB in the MTBC is independent of the secretion of these four proteins (7, 9). Another unique feature of the MTBC ESX-1 system is the requirement of EspA, EspB, EspC, EsxA, and EsxB expressions for the entire secretion system to function (1, 5, 7). Finally, while EspD is also needed for ESX-1 function, its export has been shown to be mostly ESX-1 independent (5, 16). Based on our current understanding, the minimal functioning unit of the MTBC ESX-1 system consisting of the translocon core, associated chaperones, ATPases, and secreted effectors is all encoded by genes in the *esx-1* locus (*espE/rv3864* to *mycP₁/rv3883c*) and the distal co-transcribed *espACD* operon (*espA/rv3616c, espC/rv3615c,* and *espD/rv3614c*) in the MTBC (1, 5, 8). While much has been learned about the MTBC ESX-1 system, additional components of the system and regulators of its activity have yet to be identified. Moreover, gaps in our understanding of its spatiotemporal architecture and activity also remain. Major impediments to closing these gaps include the slow growth rate and the need for biocontainment level 3 facilities to study the ESX-1 system in the MTBC.

Given these challenges, orthologous ESX-1 systems of other mycobacterial species with lower biocontainment requirements have been studied in order to understand how the MTBC ESX-1 system works. These include the ESX-1 systems of *Mycobacterium marinum*, a fast-growing pathogen that causes a TB-like disease in cold-blooded vertebrates (17) and *Mycobacterium smegmatis*, a non-pathogenic and rapidly growing model organism with a robust track record in mycobacteria research (18, 19). Although valuable mechanistic insights have been gained, there are notable differences between the ESX-1 systems of these surrogate mycobacteria and that of the MTBC that must be taken into account and not extrapolated to the MTBC ESX-1 (1, 9, 17, 19). Indeed, unlike the MTBC, *M. smegmatis* employs its ESX-1 system for conjugative DNA transfer and not for virulence (1, 20). These differences notwithstanding, individual *M. tb* genes encoding ESX-1 proteins have been expressed in *M. smegmatis* for the purpose of purifying and characterizing the recombinant *M. tb* ESX-1 proteins (21), while in others, recombinant *M. smegmatis* expressing individual ESX-1 proteins like EsxA and EsxB, and ESX-1-associated proteins like EspC, have been constructed and their interactions with a host interrogated (22, 23).

In this study, genes encoding known proteins of the *M. tb* ESX-1 system were transformed into the *M. smegmatis* $mc^2$-155 strain which is used widely in mycobacterial research (19). Thus, *M. smegmatis* strains with DNA containing the *M. tb esx-1* locus alone, *espACD* operon alone, or both the *esx-1* locus and *espACD* operon were constructed and characterized. Our findings underscore the utility of using *M. smegmatis* as a chassis to study the *M. tb* ESX-1 system.

## MATERIALS AND METHODS

### Enzymes and reagents

DNA modifying enzymes and high-fidelity Phusion DNA polymerase were purchased from New England Biolabs (Ipswich, MA, USA). Oligonucleotides were purchased from Integrated DNA Technologies (Coralville, IA, USA). All other chemicals and reagents used were purchased from Sigma-Aldrich (Oakville, ON, Canada).

### Bacterial strains and growth conditions

*M. tuberculosis* Erdman (*M. tb*) (5–7, 9, 12), *Mycobacterium bovis* BCG Pasteur 1173P2 (BCG) (24), *M. smegmatis* $mc^2$-155 (18, 19), and recombinant *M. smegmatis* were routinely

cultured in Middlebrook 7H9 liquid media supplemented with 10% albumin-dextrose-catalase and 0.05% Tween-80 (complete 7H9 media) or on Middlebrook 7H11 agar supplemented with 10% oleic acid-albumin-dextrose-catalase (complete 7H11 agar). In addition, recombinant *M. smegmatis* was grown in the presence of 30 µg/mL kanamycin (Kan; Sigma-Aldrich, Canada) and 55 µg/mL hygromycin (Hyg; Roche, Canada).

## Construction of recombinant *M. smegmatis* strains

Recombinant *M. smegmatis* strains were obtained by transformation of DNA constructs following standard electroporation procedures (25, 26). Briefly, electrocompetent *M. smegmatis* mc$^2$-155 cells were prepared and electroporated first with the integrative cosmid 2F9 that harbors a *M. tb* DNA fragment containing 24 genes (*rv3860* to *rv3885*), which includes the *esx-1* locus (4, 11, 15, 27) to generate *M. smegmatis*::2F9. *M. smegmatis* mc$^2$-155 was also electroporated with the empty, integrative, and Hyg-resistance cassette-containing cosmid vector pYUB412 (27) on which 2F9 is based (4, 11, 15, 27) to generate *M. smegmatis*::pYUB412. Both were selected on 7H11 complete agar containing Hyg. Thereafter, *M. smegmatis*::2F9 and *M. smegmatis*::pYUB412 were transformed with an episomal Kan-resistance cassette-containing plasmid called pMD*espACD* which contains the *M. tb espACD* operon (5, 6) and selected on complete 7H11 agar containing Hyg and Kan. *M. smegmatis*::2F9 and *M. smegmatis*::pYUB412 were also transformed with the empty pMD31 plasmid on which pMD*espACD* is based and selected on complete 7H11 agar containing Hyg and Kan. Thus, four different recombinant *M. smegmatis* strains were generated: (i) *M. smegmatis*::pYUB412 + pMD31 (wild-type with empty vectors), (ii) *M. smegmatis*::pYUB412 + pMD*espACD* (with *espACD* operon only), (iii) *M. smegmatis*::2F9 + pMD31 (with *esx-1* locus only), and (iv) *M. smegmatis*::2F9 + pMD*espACD* (with *esx-1* locus and *espACD* operon), hereafter designated MSX-1.

## Construction of 2F9 Sbf1

Two Sbf1 restriction sites were identified in the 2F9 cosmid. One site was localized in a non-coding section of 2F9 and the other in the *rv3860* gene situated at the 5′ end of DNA containing the *esx-1* locus. Restriction digestion with Sbf1-HF (NEB) generated 38.25 kb and 1.2 kb linear DNA fragments which were separated by agarose gel electrophoresis. The larger DNA fragment was gel purified using the Monarch DNA gel extraction kit (NEB) and re-ligated using T4 ligase (NEB), resulting in a modified cosmid designated 2F9 Sbf1.

## Quantitative reverse transcription polymerase chain reaction analysis of *esx-1* genes

*M. smegmatis*::2F9 *and M. smegmatis*::2F9 Sbf1 strains grown to late-log phase of growth were pelleted and resuspended in RLT buffer (Qiagen). The cells were lysed using 0.1 mm zirconia beads (BioSpec Products), following which total RNA was extracted using the RNeasy Mini Kit (Qiagen) and treated with Turbo DNase (Ambion) to remove any contaminating DNA according to the Turbo DNA-free Kit protocol (Ambion). cDNA from the treated RNA was synthesized using the iScript Reverse Transcription Supermix for RT-qPCR Kit (Bio-Rad). Gene expression with gene-specific primers (Table 1) was measured with SsoAdvanced Universal SYBR Green Supermix (Bio-Rad). The gene induction ratio was normalized to MSMEG_2758 (*sigA*) rRNA, and results were analyzed using the $\Delta\Delta C_T$ method.

## *M. smegmatis* culture conditions and protein preparation for immunoblots

Growth of mycobacteria to obtain culture filtrate (CF) and cell lysate (CL) proteins for analysis was done as described previously (3, 5–7, 9, 12) with some modifications. Briefly, recombinant *M. smegmatis* were grown in complete 7H9 media containing Kan and Hyg to mid-logarithmic phase of growth, centrifuged, washed with phosphate buffer saline (PBS), and resuspended in modified 7H9 without albumin-dextrose-catalase (ADC) and

**TABLE 1** List of primers used in RT-qPCR[a]

| Primers | Sequence (5′–3′) | References |
|---|---|---|
| *espB*-F and *espB*-R | GGGAACATCCGACTTATGAAGA and CTTGTTGTTGTATTCGGTCAGC | (28) |
| *esxB*-F and *esxB*-R | GCAGGAGGCAGGTAATTTCG and CCTGGTCGATCTGGGTTTTC | (29) |
| *esxA*-F and *esxA*-R | AGGGTGTCCAGCAAAAATGG and CTGCAGCGCGTTGTTCAG | (29) |
| *whiB6*-F and *whiB6*-R | CGCGGCAGAGGCTACAAC and GGCGGTTACTGTCATGTCTACGT | (29) |
| *sigA* (MSMEG_2758)-F and *sigA* (MSMEG_2758)-R | GACTACACCAAGGGCTACAAG and TTGATCACCTCGACCATGTG | (30) |
| *pe35*-F and *pe35*-R | AAGTGAGCGACAACGCTCTG and TCGTCGATTTGCGAATAGGT | (31) |
| *ppe68*-F and *ppe68*-R | GCTGATGTCTCAGCTGATCG and GTCGTCTTCACGCTCCT | (31) |
| *rv3860*-F and *rv3860*-R | TCACCATTTCGCAATCTCGG and GCCATAATCCGACAGCCATT | This study |
| *rv3861*-F and *rv3861*-R | GTCGGCAACAGCAGGATC and TGTTCCGCAGACCCCTCGAA | This study |

[a] F, forward primer; R, reverse primer (Table 1).

Tween-80 (m7H9) but containing Kan and Hyg at a starting $OD_{600nm}$ of 0.2 and cultured at 37°C with agitation. Aliquots of cell cultures were then centrifuged at the indicated times to obtain culture supernatants and bacterial pellets. Culture supernatants were then concentrated in Vivaspin columns (Sartorius, Canada) with 5 kDa molecular weight cutoff membranes to obtain CFs. CLs were prepared by resuspending bacterial pellets in PBS containing EDTA-free protease inhibitor cocktail tablets (Roche, Canada), bead beating using 100 μm glass beads, and clarification by centrifugation. Total protein concentrations in the CFs and CLs of the different mycobacterial strains were determined using the bicinchoninic acid (BCA) assay (Thermo Fisher, Canada) with bovine serum albumin as the standard.

## Immunoblotting

Immunoblotting was done as described previously (3, 5–7, 9, 12) with slight modifications. Briefly, indicated amounts of total CF and CL proteins per well were separated in NuPAGE 4-12% Bis-Tris gels (Thermo Fisher, Canada) and transferred to nitrocellulose membranes using the iBlot2 system (Thermo Fisher, Canada). Membranes were blocked with tris buffered saline (TBS)-milk (20 mM Tris-HCl, pH 7.5, 150 mM NaCl and 5% non-fat milk powder) and incubated overnight with primary antibody diluted in tris buffered saline with tween-20 (TNT-BSA) (20 mM Tris-HCl, pH 7.5, 150 mM NaCl, 0.05% Tween-20 and 1% BSA fraction V) at 4°C. Membranes were washed with TNT, incubated with the appropriate fluorescent secondary antibody in TNT-BSA, washed three times with TNT, and scanned using the Odyssey CLx Imaging system (Li-Cor Biosciences, Canada). GroEL2 was used as a lysis control for CF and as a loading control for CL. ESX-1-independent secreted protein Ag85 was probed as a loading control for CF. Rat polyclonal antibodies to EspB and EspD produced by Eurogentec S.A. for our lab were used at dilutions of 1:1,000. Commercial mouse monoclonal antibody to EsxA (Abcam), rabbit polyclonal antibody to EsxB (BEI Resources), mouse monoclonal antibody to GroEL2 (BEI Resources), and rabbit polyclonal antibody to Ag85 (BEI Resources) were used at dilutions of 1:2,000.

### *In vitro* growth measurement of recombinant *M. smegmatis* strains

Each of the recombinant *M. smegmatis* strains was inoculated into flasks containing 7H9 complete media with Kan and Hyg at a starting $OD_{600nm}$ of 0.2 and grown at 37°C with agitation. Aliquots were removed for $OD_{600nm}$ measurements at the different indicated time points. At least two independent growth experiments were conducted.

## THP-1 assays

Human THP-1 monocyte cells (TIB-202, ATCC) were cultured, differentiated using phorbol ester (PMA) into macrophage-like cells, and seeded in 96-well tissue culture plates as described previously (3, 6, 7, 9). Recombinant *M. smegmatis* strains for THP-1 infections were cultured in 7H9 complete with Kan and Hyg. After measuring $OD_{600nm}$ and calculating the colony-forming units per milliliter (CFU/mL), aliquots of the mycobacteria

were added to RPMI media supplemented with 10% fetal bovine serum to obtain the required CFU/mL for the desired multiplicities of infection (MOIs).

To quantify cytotoxicity induced in macrophages, mycobacteria suspended in RPMI were used to infect THP-1 cells in 96-well plates at an MOI of 1. Survival of THP-1 cells 12 h post-infection was quantified using the PrestoBlue reagent (Thermo Fisher, Canada) as described previously (3, 6, 7, 9).

To determine the intracellular replication of different mycobacteria in THP-1 cells, mycobacteria suspended in RPMI were used to infect THP-1 cells in 24-well plates at an MOI of 0.25. After 2 h, the media in each well of infected cells were replaced with fresh complete RPMI containing 10 µg/mL gentamicin. At different time points post-infection (2 h, 10 h, 24 h and 48 h), cells in corresponding wells were washed once with PBS and then lysed with PBS containing 0.1% Triton X-100. The resulting THP-1 cell lysates were serially diluted and plated on 7H10 medium supplemented with oleic acid-albumin-dextrose-catalase (OADC) plus Kan and Hyg to recover mycobacteria for CFU counting.

## Protective efficacy trial in the mouse model of TB

Six-week-old female C57BL/6 mice were randomly assigned to five groups of eight mice each, and mice in each group were vaccinated subcutaneously once with $1 \times 10^6$ CFU/animal of either *M. smegmatis*::pYUB412 + pMD31, *M. smegmatis*::2F9 + pMD31, MSX-1, *M. bovis* BCG Pasteur 1173P2, or saline. Three weeks post-vaccination, two animals per vaccine group were sacrificed, and spleens were collected for splenocyte assays to measure *M. bovis* antigen-specific T cell-mediated interferon-gamma (IFN-γ) production (see procedure below). The remaining mice were then challenged via the intranasal route with $1 \times 10^3$ CFU/animal of virulent *M. tb* Erdman, monitored daily, and euthanized after 28 days. Spleens and lungs from all animals were collected, homogenized, and plated out on 7H11 complete agar to enumerate *M. tb* burden in these organs (four mice per vaccine group) and for histopathological analysis (two mice per vaccine group).

## Splenocyte assay

This assay was performed as described previously (15). Briefly, spleens collected from the different groups of vaccinated mice were macerated, and the isolated splenocytes were cultured in 96-well plates at $1 \times 10^6$ cells/well in duplicate wells. Splenocytes from each mouse per vaccination group were incubated with either RPMI tissue culture media, 5 µg/mL purified protein derivative (PPD) (bovine tuberculin PPD 3000, Prionics Lelystad B.V.), 5 µg/mL *M. bovis* BCG Pasteur CF (prepared in-house), and 1 µg/mL concanavalin A or ConA (Sigma-Aldrich, Canada) for 72 h at 37°C and 5% $CO_2$. The IFN-γ in splenocyte culture supernatant was quantified using the Invitrogen IFN-γ mouse ELISA kit (Thermo Fisher, Canada) according to the manufacturer's instructions.

## Histopathology

Lung tissue samples from the different groups of vaccinated mice were held in 10% normal buffered formalin for about 1 week and embedded in paraffin, and sections were stained with hematoxylin and eosin (H & E) to assess inflammation as described previously (32).

## Statistical analysis

GraphPad Prism was used to plot quantitative data and for statistical analyses as indicated. Analysis of differences in cytotoxicity caused by *M. smegmatis* in THP-1 macrophage cells was done using one-way analysis of variance (ANOVA) with Tukey's multiple comparisons. Analysis of differences in intracellular burdens of *M. smegmatis* in THP-1 macrophage cells was done using two-way ANOVA with Tukey's multiple comparisons. Analysis of differences in transcription in RT-qPCR experiments was done using Student's *t*-test (two-tailed, unpaired, parametric).

## RESULTS

### *M. smegmatis* transformed with genes of the *M. tb esx-1* locus and *espACD* operon expresses an optimally functioning *M. tb* ESX-1 system

*M. smegmatis* mc$^2$-155 was transformed with the integrating 2F9 cosmid containing a fragment of *M. tb* genomic DNA encompassing the *esx-1* locus, the episomal pMD*espACD* plasmid which contains the *espACD* operon under native promoter control (Fig. 1A), and the corresponding empty vectors pYUB412 and pMD31 to generate *M. smegmatis*::pYUB412 + pMD31 (wild-type), *M. smegmatis*::pYUB412 + pMD*espACD*, *M. smegmatis*::2F9 + pMD31, and *M. smegmatis*::2F9 + pMD*espACD* (designated MSX-1). Typically, MTBC secretome analysis is done using culture supernatants of the organism grown in Sauton's media, a chemically defined liquid media (9). However, *M. smegmatis,* which has an orthologous ESX-1 system but not an orthologous *espACD* operon (Table 2), was reported to export its own ESX-1 protein substrates in Sauton's media but not in a modified 7H9 liquid media lacking ADC and Tween-80 (33). Therefore, to differentiate between *M. tb* and *M. smegmatis* ESX-1-secreted proteins, we performed our immunoblot analysis on protein fractions from culture supernatants and cell pellets of recombinant *M. smegmatis* grown in modified 7H9 liquid media lacking ADC and Tween-80. Accordingly, *M. tb* EsxA and EsxB were not detected in 5 µg of the CF and 2.5 µg of CL protein fractions of *M. smegmatis*::pYUB412 + pMD31 or *M. smegmatis*::pYUB412 + pMD*espACD* using antibodies specific to these proteins (Fig. 1B and C). And while *M. tb* EsxA and EsxB were detected in the CF and CL fractions of both *M. smegmatis*::2F9 + pMD31 and *M. smegmatis*::2F9 + pMD*espACD* (or MSX-1), their signals were stronger in the CF and CL fractions of MSX-1 (Fig. 1B and C). Likewise, using antibodies specific to *M. tb* EspB, this protein was only detected in the CF and CL fractions of *M. smegmatis*::2F9 + pMD31 and MSX-1, although the signal was stronger in the latter (Fig. 1B and C). While *M. tb* EspD was detected in the CL fraction of *M. smegmatis*::pYUB412 + pMD*espACD*, the amount of this protein in the CL fraction of MSX-1 was noticeably higher (Fig. 1C). Moreover, EspD was only detected in the CF fraction of MSX-1 after 20 and 30 h of culture (Fig. 1B). SDS-PAGE separation and immunoblotting of higher amounts of CF (15 µg/well) and CL (7.5 µg/well) proteins confirmed that the observed differences in the levels of ESX-1-associated proteins are reproducible (Fig. S1A and B).

The differences in the levels of EspB, EspD, EsxA, and EsxB in the CF fractions noted above were not due to unequal total CF protein loading as the signal for Ag85, an ESX-1-independent secreted protein (3), was similar in the CF fractions of all four strains at the corresponding time points (Fig. 1B). Likewise, differences in the abundance of EspB, EspD, EsxA, and EsxB in the CL fractions described were likely not due to unequal loading of total CL proteins, since the signals for GroEL2, a cell-associated chaperonin (3), and Ag85 were similar in the CL fractions of all four strains at corresponding time points (Fig. 1C). The absence of GroEL2 in the CF fractions of all four strains indicates that the presence of EsxA, EsxB, EspB, and EspD in some was not due to cell autolysis (Fig. 1B).

### *M. smegmatis* producing the *M. tb* ESX-1 system grow at similar rates

To determine if differences in growth rates among the *M. smegmatis* strains might be the reason for their variable ESX-1 protein expression and secretion, we compared the growth of *M. smegmatis*::pYUB412 + pMD31, *M. smegmatis*::pYUB412 + pMD*espACD*, *M. smegmatis*::2F9 + pMD31, and MSX-1 in 7H9 media supplemented with Kan and Hyg (Fig. 2). All four recombinant *M. smegmatis* strains were observed to grow at the same rate in this liquid culture media.

### *M. smegmatis* producing *M. tb* ESX-1 is non-virulent in the macrophage model of infection

Wild-type *M. smegmatis* is non-pathogenic and non-cytotoxic toward mammalian cells (19). However, as the ESX-1 system is critical to *M. tb* virulence and ESX-1 mediated export of EsxA, EsxB, EspA, EspB, and EspC is needed by *M. tb* to induce cell death

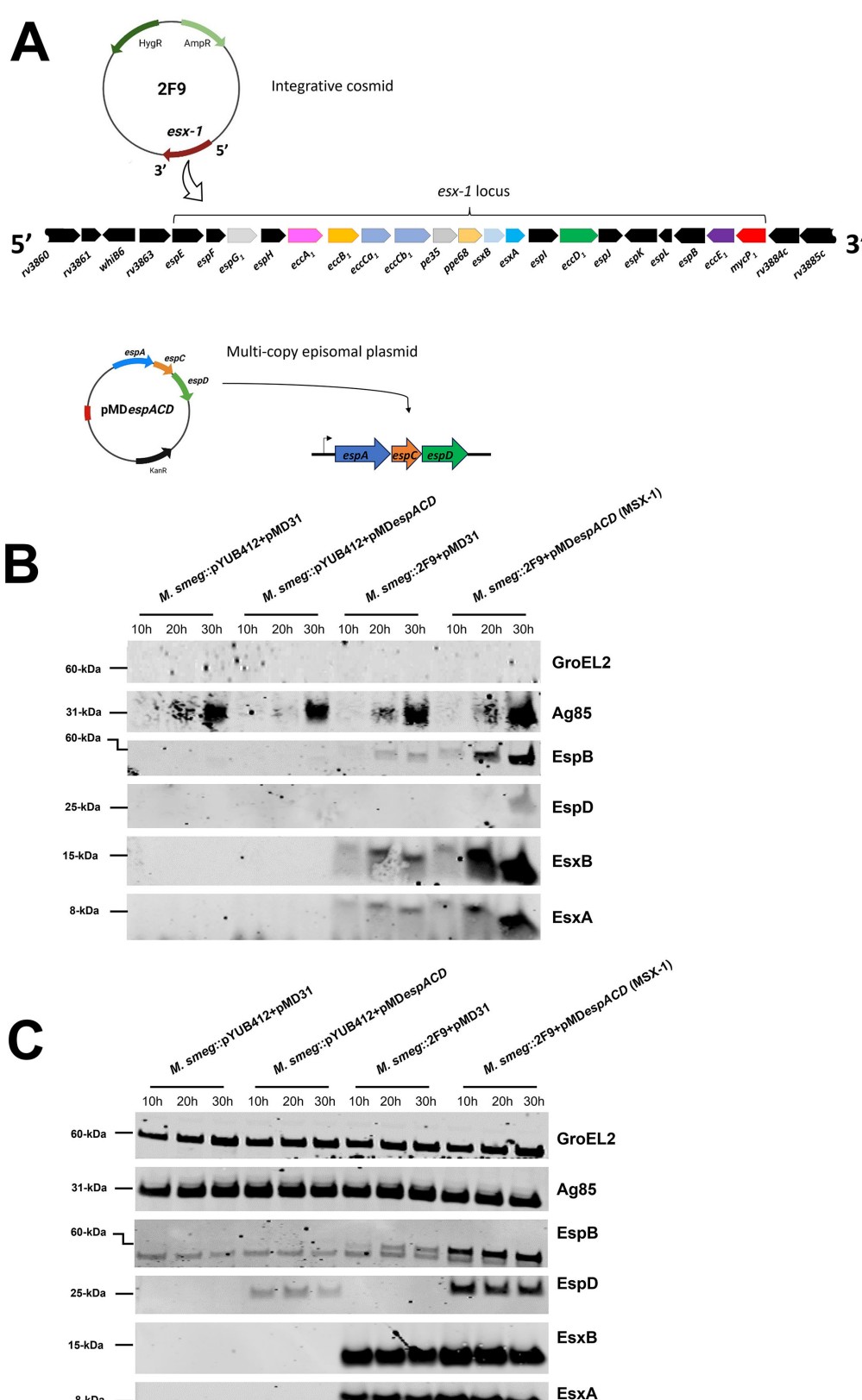

FIG 1   Immunoblot analysis of recombinant *M. smegmatis* strains. (A) Schematic of the 2F9 integrative cosmid containing genes of the *esx-1* locus (starting at a truncated *rv3860* and ending at a truncated *rv3885c*) and pMD*espACD* plasmid containing the *espACD* operon under control of its native promoter. Immunoblots of (B) CF proteins and (C) CL proteins of *M.*

**Fig 1 (Continued)**

*smegmatis*::pYUB412 + pMD31, *M. smegmatis*::pYUB412 + pMD*espACD*, *M. smegmatis*::2F9 + pMD31, and *M. smegmatis*::2F9 + pMD*espACD* (MSX-1) after 10, 20, and 30 h of growth in modified 7H9 media. Immunoblots shown are representative of at least three independent experiments.

in macrophages (3, 6, 7, 9), we wanted to know if *M. smegmatis* strains expressing *M. tb* ESX-1 might make them cytotoxic to macrophages. Accordingly, human THP-1 macrophage cells were co-cultured with either *M. smegmatis*::pYUB412 + pMD31, *M. smegmatis*::2F9 + pMD31, MSX-1, or left untreated, and the viability of the macrophages was assessed after 12 h. Compared to uninfected THP-1 cells, none of the cells infected with the recombinant *M. smegmatis* strains were observed to show reduced viability (Fig. 3A).

Given that *M. tb* requires a fully functional ESX-1 system to replicate in macrophages (34–36), while wild-type *M. smegmatis* fails to replicate and is eventually cleared after being taken up by macrophages (37, 38), we examined the fate of the recombinant *M. smegmatis* strains in THP-1 cells. To this end, *M. smegmatis*::pYUB412 + pMD31, *M. smegmatis*::2F9 + pMD31, and MSX-1 were individually co-cultured with THP-1 cells, and intracellular mycobacteria were recovered and quantified at different time points. No differences in the numbers of *M. smegmatis*::pYUB412 + pMD31, *M. smegmatis*::2F9 + pMD31, and MSX-1 taken up by the THP-1 cells after 2 h of co-culture were observed (Fig. 3B). Relative to the number of intracellular mycobacterial cells at 2 h post-infection, the number of intracellular *M. smegmatis*::pYUB412 + pMD31, *M. smegmatis*::2F9 + pMD31, and MSX-1 all decreased by similar amounts at 10, 24, and 48 h post-infection (Fig. 3C).

To determine whether the lack of cytotoxicity and unenhanced intracellular persistence of MSX-1 might be due to its loss of the episomal pMD*espACD* in THP-1 cells, we assessed EspD expression in MSX-1 grown in media with and without Kan over increasing lengths of time as a way to gauge maintenance of pMD*espACD*. Indeed, EspD

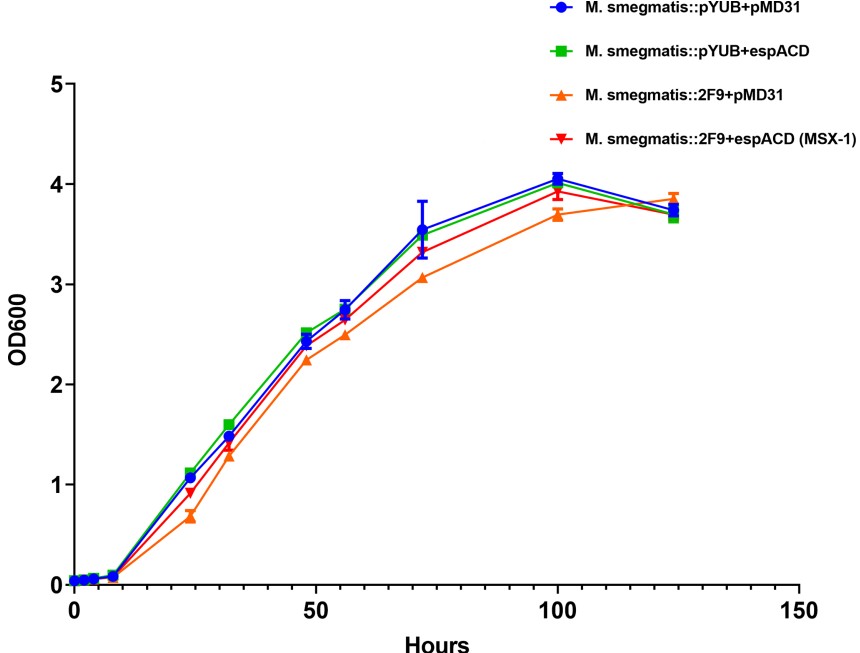

**FIG 2** Growth comparison of recombinant *M. smegmatis* strains. *M. smegmatis*::pYUB412 + pMD31, *M. smegmatis*::pYUB412 + pMD*espACD*, *M. smegmatis*::2F9 + pMD31, and *M. smegmatis*::2F9 + pMD*espACD* (MSX-1) were cultured in 7H9 with Hyg and Kan, and OD$_{600nm}$ readings taken at the indicated time points. Each data point is the means, and error bars are the standard deviation of the OD$_{600nm}$ values of two independent experiments.

levels in MSX-1 grown in the absence of Kan for up to 8 days were comparable to what is seen in the same strain grown in the presence of the antibiotic over the same length of time (Fig. S2). This strongly suggests the plasmid construct is maintained and not rapidly lost by MSX-1 upon withdrawal of Kan selection pressure. As such, pMD*espACD* loss is likely not the mechanism underlying its unaltered cytotoxicity and intracellular persistence in macrophage cells.

## *M. smegmatis* transformed with a modified 2F9 cosmid reveals *rv3860* to be associated with the transcriptional activation of *pe35*, *ppe68*, *esxB,* and *esxA*

We next sought to subclone the *espACD* operon into the integrative 2F9 cosmid for subsequent transformation into *M. smegmatis* and dispense with having to select for the episomal plasmid using Kan. Accordingly, two unique Sbf1 restriction sites in 2F9 were identified—one in a non-coding section of 2F9 and the other in the *rv3860* gene situated at the 5′ end of DNA containing the *esx-1* locus (Fig. 4A). In the annotated *M. tb* H37Rv genome, *rv3860* is predicted to be a gene of 1,173 bp (39, 40). In 2F9, however, 27 bp of *rv3860,* including its predicted start codon, is missing (Fig. 4A). As such, digestion with Sbf1 and its religation to yield 2F9 Sbf1 would remove some of the 2F9 cosmid and about 1 kb of *rv3860* DNA (Fig. 4A) and is not expected to impact the *esx-1* locus. Indeed, sequencing of the resulting 2F9 Sbf1 cosmid showed no other difference when compared to 2F9. Nevertheless, to verify that 2F9 and 2F9 Sbf1 function similarly, both were transformed into *M. smegmatis* to generate *M. smegmatis*::2F9 and *M. smegmatis*::2F9 Sbf1, respectively. Surprisingly, immunoblot analysis of the CL of *M. smegmatis*::2F9 and *M. smegmatis*::2F9 Sbf1 for *M. tb* EsxA and EsxB revealed the total absence of these proteins in the latter strain (Fig. 4B).

   *M. tb* is 99.95% genetically identical to *M. bovis*, another MTBC member and a zoonotic agent of bovine TB (41). The live attenuated TB vaccine *M. bovis* BCG was derived from a virulent strain of *M. bovis* through *in vitro* passaging, and a portion of its *esx-1* locus from *eccC_{b1}* to *espK* was lost in the process (24). As a consequence, BCG does not make EsxA or EsxB; however, this can be restored upon transformation with 2F9 (11, 15). To determine if the modified 2F9 can restore EsxA and EsxB production in the vaccine strain, *M. bovis* BCG Pasteur was transformed with 2F9 Sbf1 as well as with 2F9 for comparison. CLs of the resulting Pasteur::2F9 and Pasteur::2F9 Sbf1 strains were then immunoblotted for *M. tb* EsxA and EsxB. Unlike *M. smegmatis,* however, both BCG strains were found to make EsxA and EsxB (Fig. 4C).

   Examination of the truncated *rv3860* sequence in 2F9 revealed the presence of multiple alternative start codons encoding methionine, valine, and even leucine, which is a hallmark of the mycobacterial genetic code (42). Also, it has been shown that *pe35*, *ppe68*, *esxB,* and *esxA* genes are co-expressed in *M. tb* (5, 7, 9). We therefore hypothesized that despite the 27 bp truncation, *rv3860* may yet be expressed off 2F9 to produce a functional protein with a role to play in the expression of *M. tb pe35*, *ppe68*, *esxB,* and *esxA* genes. To test this notion, we first assessed if *rv3860* is transcribed in *M. smegmatis*::2F9. Thereafter, the transcription of *M. tb rv3861*, *whiB6*, *pe35*, *ppe68*, *esxB*, *esxA,* and *espB* in *M. smegmatis*::2F9 and *M. smegmatis*::2F9 Sbf1 was examined and compared by quantitative reverse transcription polymerase chain reaction (RT-qPCR). We chose to examine *rv3861* because its gene sequence overlaps and is potentially co-transcribed with *rv3860* (40). We also examined *whiB6* and *espB* as these genes are on the antisense strand and transcribed in the opposite direction to *rv3860*, *pe35*, *ppe68*, *esxB,* and *esxA* (40). Strikingly, our RT-qPCR results showed that *rv3860* is indeed transcribed in *M. smegmatis*::2F9 (Fig. 4D). More notably, when compared to *M. smegmatis*::2F9, transcription of *pe35*, *ppe68*, *esxB,* and *esxA,* but not *rv3861*, *whiB6*, or *espB,* was found to be significantly reduced in *M. smegmatis*::2F9 Sbf1 (Fig. 4D).

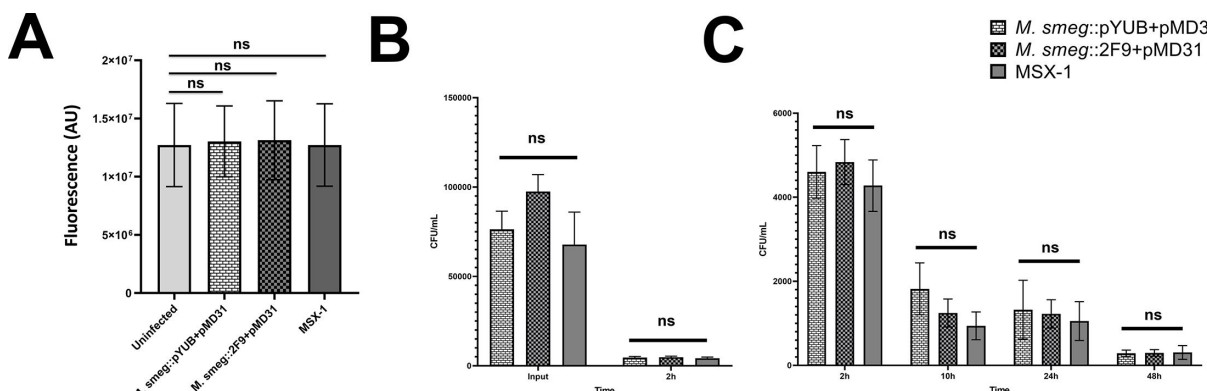

**FIG 3** THP-1 cell infections with recombinant *M. smegmatis* strains. (A) Assessment of cytotoxicity caused by *M. smegmatis*::pYUB412 + pMD31, *M. smegmatis*::2F9 + pMD31, and *M. smegmatis*::2F9 + pMD*espACD* (MSX-1) in THP-1 macrophage cells infected at an MOI of 1 and measured using PrestoBlue reagent. Data shown are the mean total fluorescence signal in arbitrary units from viable THP-1 cells with error bars representing standard deviations of three independent experiments. Statistical significance of differences in results was determined by one-way ANOVA, followed by Tukey's multiple comparison test. (B) Intracellular burdens of *M. smegmatis*::pYUB412 + pMD31, *M. smegmatis*::2F9 + pMD31, and *M. smegmatis*::2F9 + pMD*espACD* (MSX-1) in macrophage-like THP-1 cells infected at an MOI of 0.25 after 2 h compared to corresponding input mycobacteria. (C) Intracellular burdens of *M. smegmatis*::pYUB412 + pMD31, *M. smegmatis*::2F9 + pMD31, and *M. smegmatis*::2F9 + pMD*espACD* (MSX-1) in macrophage-like THP-1 cells infected at an MOI of 0.25 after 2, 10, 24, and 48 h post-infection. Data shown are mean CFU/mL with error bars representing standard deviations from THP-1 cells of two independent experiments. ns, not significant. Two-way ANOVA followed by Tukey's multiple comparisons was used to determine the statistical significance of results.

## *M. smegmatis::* 2F9 + pMD *espACD* (MSX-1) is as protective as the live attenuated *M. bovis* BCG vaccine against *M. tb* infection

Given that the recombinant *M. smegmatis* strains we constructed were found to be non-cytotoxic and several of the secreted *M. tb* ESX-1 proteins are well-known T cell antigens (4, 13–15), we wanted to examine their traits in an animal model and if they might provide protection against subsequent *M. tb* infection, especially in comparison to *M. bovis* BCG, the only licensed TB vaccine available to date (24). Accordingly, mice were subcutaneously injected with either *M. smegmatis*::pYUB412 + pMD31, *M. smegmatis*::2F9 + pMD31, MSX-1, or BCG, and monitored. Having observed no adverse effects in vaccinated mice, namely reduced grooming, morbidity, inappetence, and weight loss by 3 weeks post-injection, two mice from each group were euthanized, and their spleens were removed to isolate splenocytes for tuberculin sensitivity testing. The remaining mice were challenged with virulent *M. tb*. Approximately 1 month post-challenge, the mice were euthanized and the *M. tb* burden in their lungs and spleens was quantified. We noted that the median *M. tb* burden was about 1 log lower in the lungs of mice vaccinated with either MSX-1 ($P < 0.05$, one-way ANOVA with Tukey's multiple comparison test) or BCG ($P < 0.005$, one-way ANOVA with Tukey's multiple comparison test) than in the lungs of mock-vaccinated mice and in mice vaccinated with *M. smegmatis*::pYUB412 + pMD31 and *M. smegmatis*::2F9 + pMD31 (Fig. 5A). Likewise, the median *M. tb* burden in the spleens of mice vaccinated with either MSX-1 or BCG was about half-log lower than the burden in the spleens of mock-vaccinated mice and in mice administered *M. smegmatis*::pYUB412 + pMD31 or *M. smegmatis*::2F9 + pMD31, although the differences were not statistically significant (Fig. 5B).

Hematoxylin-eosin staining of fixed lung tissue slices revealed some inflammation and what appeared to be multifocal neutrophilic infiltration and thickening of the alveolar septa in the lungs of *M. tb*-challenged mice previously vaccinated with *M. smegmatis*::pYUB412 + pMD31, which is consistent with TB-mediated pathology (Fig. 6A). In mice administered either MSX-1 or BCG, the lungs presented with normal architecture and limited inflammation (Fig. 6B and C). In contrast, lungs of mock-vaccinated mice presented with extensive inflammation as well as necrotic regions surrounded by

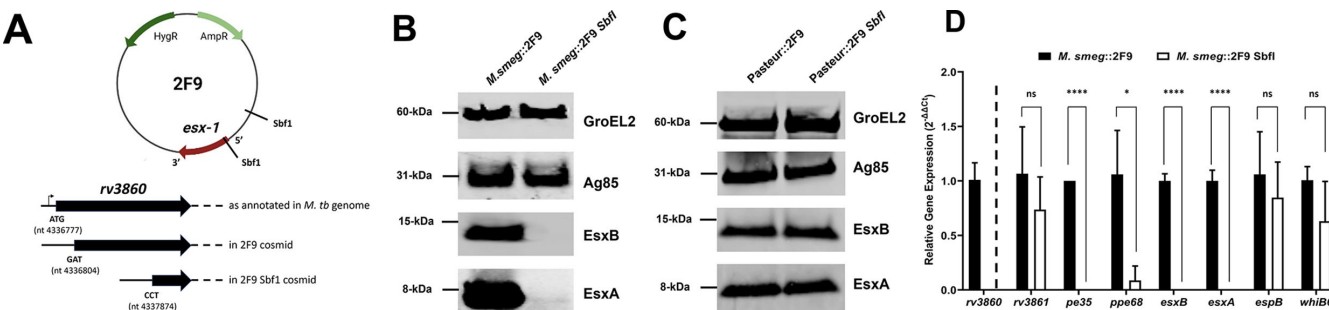

**FIG 4** Impact of *rv3860* in 2F9 transformed into *M. smegmatis* and BCG and its effects on transcription of *esx-1* locus genes. (A) Schematic of Sbf1 restriction sites in 2F9 and the *rv3860* gene as annotated in the *M. tb* H37Rv genome, 2F9 and 2F9 Sbf1. (B) Immunoblot analysis of *M. smegmatis*::2F9 and *M. smegmatis*::2F9 Sbf1 cell lysate proteins. (C) Immunoblot analysis of *M. bovis* BCG Pasteur::2F9 and *M. bovis* BCG Pasteur::2F9 Sbf1 cell lysate proteins. Five micrograms per well of proteins was loaded and separated by SDS-PAGE before transferring to nitrocellulose membranes. Blots are representative of at least two independent experiments. (D) Transcription of *rv3860* in *M. smegmatis*::2F9 and comparative transcription of *rv3860*, *pe35*, *ppe68*, *esxB*, *esxA*, *espB*, and *whiB6* in *M. smegmatis*::2F9 vs *M. smegmatis*::2F9 Sbf1. Transcripts are normalized relative to *M. smegmatis sigA* and plotted as mean ± SD of three independent biological experiments. ****$P < 0.0001$; *$P < 0.05$; ns, not significant. Student's *t*-test (unpaired, parametric) was used to determine the statistical significance of differences in results.

massive infiltration of neutrophils and significant loss of lung architecture, consistent with TB-mediated pathology (Fig. 6D).

## *M. smegmatis::* 2F9 + pMD *espACD* or MSX-1 does not sensitize a vaccinated host to tuberculin

BCG vaccination sensitizes the host to *M. tb* complex-specific antigens via delayed-type hypersensitivity (DTH) (4, 11, 15, 43). This is manifested by the production of IFN-γ by primed T cells of the vaccinated host upon stimulation with tuberculin or PPD, a complex mixture of MTBC-secreted proteins (4, 11, 15, 43). We sought to determine if vaccination with the recombinant *M. smegmatis* strains would have the same effect. Therefore, splenocytes, which are enriched in primed T cells, were isolated from the spleens of mock-vaccinated and vaccinated mice and stimulated with either RPMI media alone (negative control), bovine PPD (tuberculin from *M. bovis*), BCG CF, or ConA, and the amount of IFN-γ induced was quantified by ELISA. As expected, RPMI media did not stimulate splenocytes from any of the mice, while ConA, a non-specific stimulator of T cells (4, 11, 15), induced high levels of IFN-γ production in splenocytes from all mice (Fig. 7). In contrast, BCG CF was found to stimulate IFN-γ production in splenocytes from mice administered BCG but not in splenocytes from mice vaccinated with either *M. smegmatis*::pYUB412 + pMD31, *M. smegmatis*::2F9 + pMD31, or MSX-1 (Fig. 7). Likewise, bovine PPD was found to stimulate IFN-γ production in splenocytes only from BCG-vaccinated mice but not in splenocytes from mice vaccinated with the three *M. smegmatis*-based strains (Fig. 7).

## DISCUSSION

It is well established that the key genes encoding proteins of the *M. tb* ESX-1 system are in the *esx-1* locus and the *espACD* operon (1, 4, 5, 8, 9, 12). In this study, we wanted to determine if introducing these genes into the non-pathogenic, rapidly growing, and genetically tractable *M. smegmatis* might lead to the assembly of a functional *M. tb* ESX-1 in this surrogate mycobacterium.

Our immunoblot results show that transformation with genes in both the *M. tb esx-1* locus and *espACD* operon in MSX-1 results in higher levels of key *M. tb* ESX-1 proteins like EsxA, EsxB, EspB, and EspD being detected inside the cell of this strain and consequently in its CF fraction. While we cannot be certain if the high levels of EsxA, EsxB, and EspB in MSX-1 are due to increased expression and/or increased stability, it is consistent with *espACD* genes being essential for *M. tb* ESX-1 function (3, 5, 6, 8–10, 12). Likewise, the

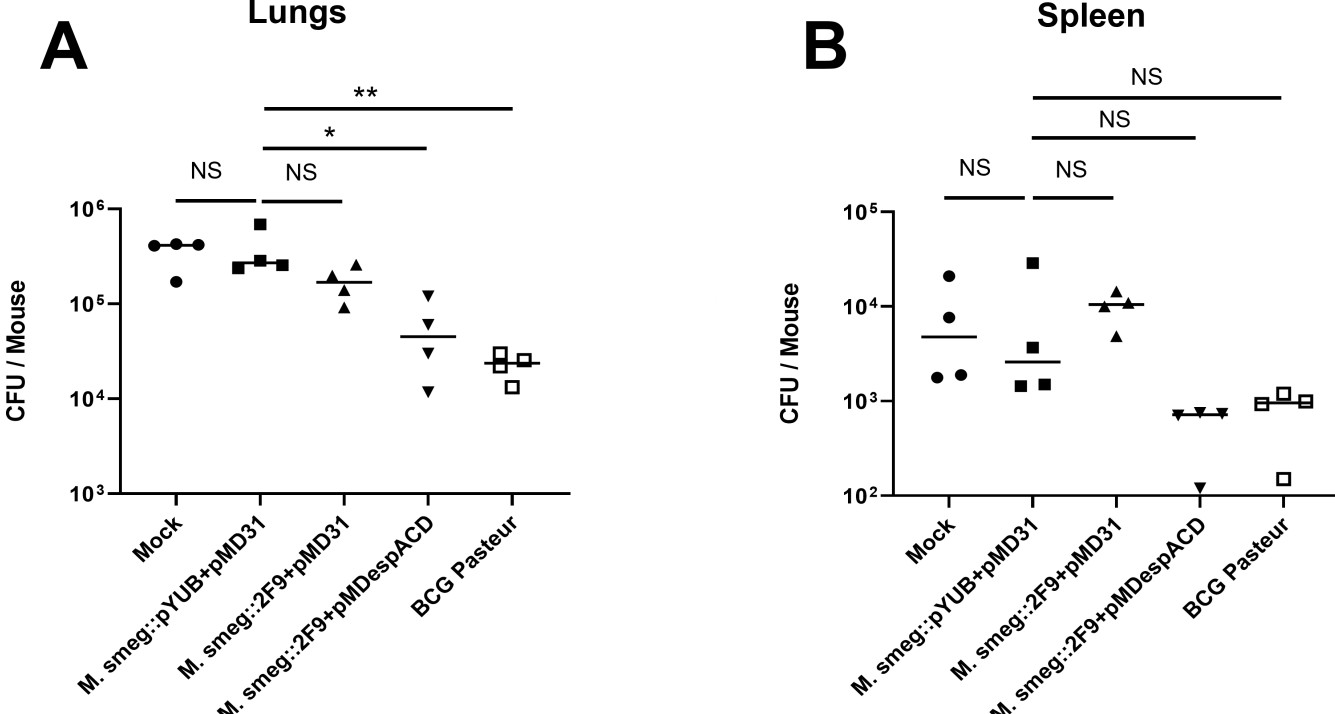

**FIG 5** *M. tb* burden in vaccinated mice. CFU of challenge *M. tb* per organ in the (A) lungs and (B) spleens of mock-vaccinated mice and mice vaccinated with either *M. smegmatis*::pYUB412 + pMD31, *M. smegmatis*::2F9 + pMD31, *M. smegmatis*::2F9 + pMD*espACD* (MSX-1), or BCG. Data point in each graph is the burden in CFU per organ from each mouse in a given treatment group. The bars represent the median CFU per organ. *$P < 0.05$; **$P < 0.005$; ns, not significant. Statistical significance of differences was determined by one-way ANOVA followed by Tukey's multiple comparison test.

higher levels of EspD observed in MSX-1 are consistent with a previous report that *M. tb* lacking *espL* has reduced levels of EspD (16). Our data do, however, confirm that the differences in the abundance of these ESX-1-associated proteins are not due to differences in growth rate of the different *M. smegmatis* strains, as none were found. Although we tested for the secretion of *M. tb* ESX-1 protein effectors by recombinant *M. smegmatis* strains in 7H9 media, which is not conducive to *M. smegmatis* ESX-1 secretion activity (33), and we used *M. tb*-specific antibodies in our immunoblot experiments, we cannot rule out the possibility that some or all of the *M. smegmatis* ESX-1 proteins may interact with *M. tb* ESX-1 proteins to form a hybrid system capable of driving the export of the *M. tb* ESX-1-secreted proteins detected in the CF fractions of *M. smegmatis*::2F9 + pMD31 and MSX-1. This is plausible, especially since *M. tb* EsxA ectopically expressed in *M. smegmatis* has been reported to be recognized and exported by the *M. smegmatis* ESX-1 system (33).

Although the *M. tb* ESX-1 system is critical for the virulence of the TB bacillus, our THP-1 macrophage infection experiments clearly show that *M. smegmatis*::2F9 + pMD31 and MSX-1 strains are neither cytotoxic nor do they exhibit enhanced intracellular survival compared to the wild-type *M. smegmatis*::pYUB412 + pMD31 strain. Our results are in contrast to the findings of Guo et al., who found that recombinant *M. smegmatis* expressing and secreting *M. tb* EspC exhibits increased persistence in mouse RAW 264.7 macrophage cells, enhanced replication in mouse lungs and spleens, and greater lethality in mice (22). This discrepancy is probably due to the marked differences between their recombinant *M. smegmatis* strain and MSX-1—the former expresses *M. tb* EspC only under the control of an IPTG-inducible promoter while MSX-1 expresses EspA, EspC, and EspD under the control of the native *espA* promoter along with other ESX-1 proteins encoded in 2F9. Guo et al. infected RAW 264.7 cells with their strain at a high MOI of 10 (22), whereas we infected THP-1 cells with MSX-1 at an MOI of 0.25 in order

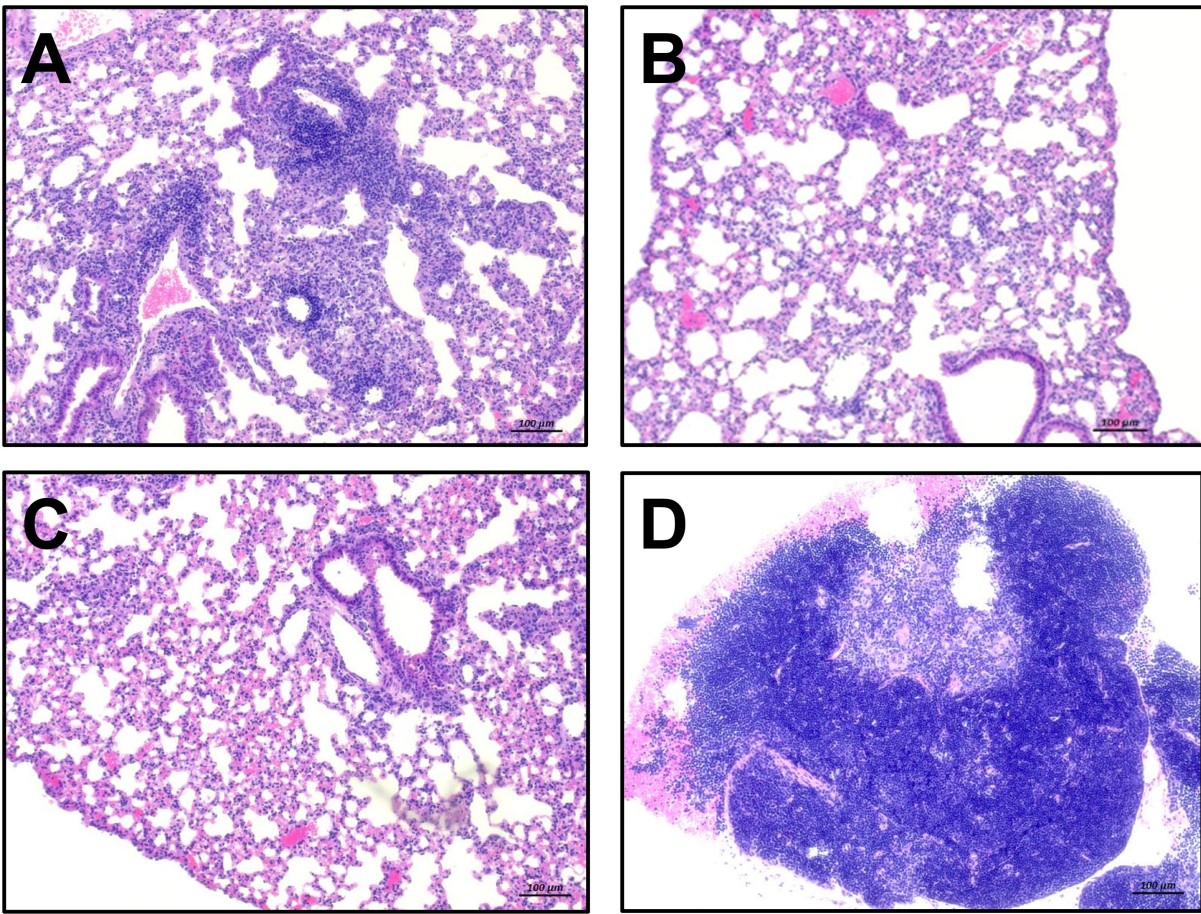

**FIG 6** *M. tb*-induced pathology in lungs of vaccinated mice. H & E staining of lung sections from *M. tb*-challenged mice vaccinated with (A) *M. smegmatis*::pYUB412 + pMD31, (B) MSX-1, (C) BCG, and (D) saline. Representative of two to three sections per mouse per vaccination group. Scale bar is 100 µm.

to measure intracellular replication. Moreover, in order to assess replication and survival in mice, Guo et al. injected $10^7$ and $10^8$ CFUs of their *M. smegmatis* strain intravenously (22), a route of *M. smegmatis* infection known to be lethal to mice (44). In contrast, we injected mice subcutaneously with $10^6$ CFUs of our *M. smegmatis* strains. Our findings are, however, consistent with the fact that in order to mediate full host-cell cytotoxicity, *M. tb* ESX-1 works in synergy with other virulence factors like the complex glycolipid phthiocerol dimycocerosates, which are not present in *M. smegmatis* (45).

Whole cosmid sequencing revealed no differences between 2F9 and 2F9 Sbf1 other than the extended truncation of *rv3860* in the latter. Nevertheless, we saw little to no EsxA and EsxB production in *M. smegmatis*::2F9 Sbf1, unlike in *M. smegmatis*::2F9, Pasteur::2F9, and Pasteur::2F9 Sbf1. We reasoned that the extended truncation of *rv3860* in 2F9 Sbf1 was the cause of the observed phenotype in *M. smegmatis* but not in BCG, as the vaccine strain has an identical version of the gene, while the *M. smegmatis* ortholog is only 49.2% identical (Table 2). Despite having a 27 bp truncation, we hypothesized that the *rv3860* gene in the 2F9 cosmid is transcribed and translated into a functional protein with a role in EsxA and EsxB expression. Indeed, this notion is supported by our observation that *M. tb rv3860* is transcribed in the *M. smegmatis*::2F9 strain. Moreover, the observation that *M. tb pe35*, *ppe68*, *esxB,* and *esxA,* but not *rv3861*, *whiB6*, and *espB,* are poorly transcribed in *M. smegmatis*::2F9 Sbf1 lends further credence to the notion that *rv3860* is involved in the transcriptional activation of the *pe35-esxA* operon. Interestingly, *rv3860,* as annotated in the *M. tb* genome, is a gene of unknown function, although it is predicted to encode a highly conserved basic protein with an isoelectric point of

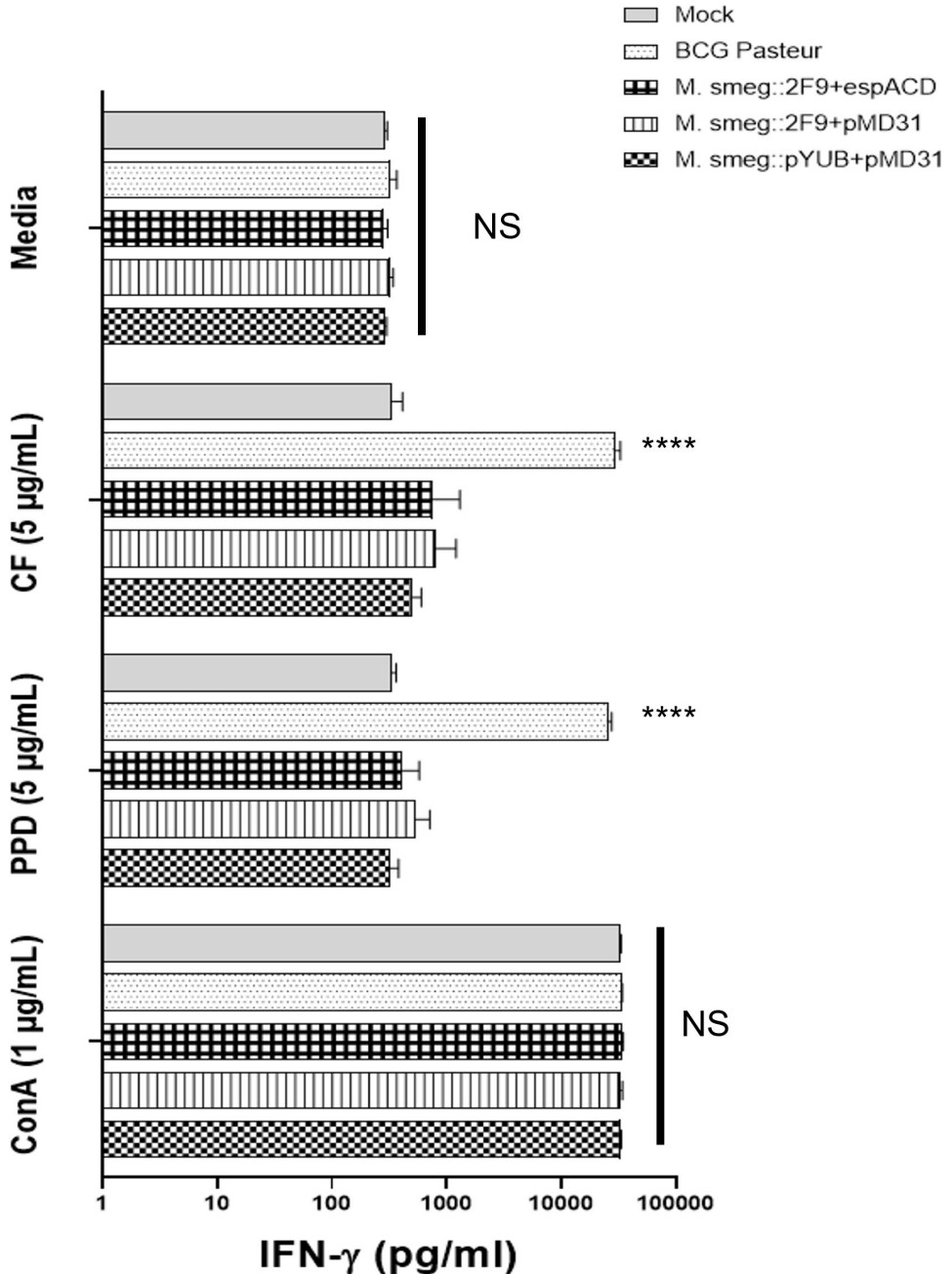

**FIG 7** Responses of splenocytes from vaccinated mice to antigens. Induction of IFN-γ by splenocytes isolated from mock-vaccinated mice and mice vaccinated with either BCG, MSX-1, *M. smegmatis*::2F9 + pMD31, or *M. smegmatis*::pYUB412 + pMD31 upon stimulation with RPMI tissue culture medium, BCG CF, bovine PPD, and ConA. Data represent the means with error bars representing the standard deviation of the concentration of IFN-γ in supernatants of splenocytes from two mice per vaccination group. ****$P < 0.0005$; ns, not significant. Statistical significance of differences was determined by one-way ANOVA, followed by Tukey's multiple comparison test for each antigen.

9.9 that is most likely localized to the cytosol of the TB bacillus (39, 40). Furthermore, its polypeptide sequence is predicted to contain a ParA-like nucleotide-binding domain (39, 40). These features suggest that the *rv3860* gene product may interact with DNA and potentially function as a transcriptional regulator. Work to address this notion and elucidate the molecular mechanism whereby *rv3860* positively regulates the transcription of *pe35-ppe68-esxB-esxA* is ongoing.

**TABLE 2** DNA sequence similarity of ESX-1 genes[a]

| M. tb H37Rv gene | M. smegmatis mc[2]-155 ortholog | Percent identity |
|---|---|---|
| rv3616c (espA) | NP[b] | NP |
| rv3615c (espC) | NP | NP |
| rv3614c (espD) | NP | NP |
| rv3860 | MSMEG_0053 | 49.20% |
| rv3861 | NP | NP |
| rv3862c (whiB6) | MSMEG_0051 | 68.97% |
| rv3863 | MSMEG_0052 | 63.22% |
| rv3864 (espE) | MSMEG_0055 | 22.06% |
| rv3865 (espF) | MSMEG_0056 | 25.00% |
| rv3866 (espG$_1$) | MSMEG_0057 | 69.45% |
| rv3867 (espH) | MSMEG_0058 | 43.27% |
| rv3868 (eccA$_1$) | MSMEG_0059 | 75.96% |
| rv3869 (eccB$_1$) | MSMEG_0060 | 65.55% |
| rv3870 (eccC$_{a1}$) | MSMEG_0061 | 80.91% |
| rv3871 (eccC$_{b1}$) | MSMEG_0062 | 77.70% |
| rv3872 (pe35) | MSMEG_0063 | 52.22% |
| rv3873 (ppe68) | MSMEG_0064 | 45.35% |
| rv3874 (esxB) | MSMEG_0065 | 62.24% |
| rv3875 (esxA) | MSMEG_0066 | 71.58% |
| rv3876 (espI) | MSMEG_0067 | 50.72% |
| rv3877 (eccD$_1$) | MSMEG_0068 | 67.00% |
| rv3878 (espJ) | MSMEG_0069 | 26.69% |
| rv3879c (espK) | MSMEG_0071 | 34.50% |
| rv3880c (espL) | MSMEG_0081 | 50.96% |
| rv3881c (espB) | MSMEG_0076 | 30.81% |
| rv3882c (eccE$_1$) | MSMEG_0082 | 66.81% |
| rv3883c (mycP$_1$) | MSMEG_0083 | 72.46% |

[a]M. smegmatis orthologs and percent DNA sequence identity to M. tb espACD and esx-1 genes sourced from https://orca1.tamu.edu/mad/orthologs/pages/Rv3860.html.
[b]NP, Not present.

The BCG-equivalent protection afforded by MSX-1 against M. tb infection is another key observation. Indeed, given that MSX-1, which harbors both the M. tb espACD operon and the esx-1 locus, produces and secretes more EsxA, EsxB, and EspB, and likely also more EspA and EspC (4, 11, 13–15), may be why it offers better protection than M. smegmatis::2F9 + pMD31. Yet another key observation is the inability of the M. smegmatis-based strains to induce DTH and sensitize the vaccinated mice to BCG CF and bovine tuberculin/PPD. One reason for the non-sensitization to BCG CF could lie in differences in the protein composition of M. smegmatis and BCG secretomes. Another possibility could be that the kinetics of DTH caused by live M. smegmatis-based strains is different from that induced by live BCG, especially considering the growth rate of M. smegmatis, which has a doubling time of approximately 3 to 4 h compared to BCG, which has a doubling time of 18 to 20 h. The lack of sensitization to bovine PPD, especially in mice administered MSX-1, is surprising given that PPDs of MTBC origin, including bovine PPD, are known to contain secreted ESX-1 proteins EsxA, EsxB, EspA, EspB, and EspC (43, 46–48). Again, differences in the kinetics of DTH induced by M. smegmatis may account for this phenotype. This observation also suggests that the nature of the protective immune response afforded by MSX-1 may be IFN-γ independent and different from that afforded by BCG. Additional studies needed to address these questions are also ongoing. Nevertheless, our results here clearly show that MSX-1 represents a novel protective TB vaccine that will potentially not compromise existing TB diagnostic tests.

## ACKNOWLEDGMENTS

We wish to thank the VIDO Veterinary Services group for their technical assistance in carrying out the mouse trial.

This work was supported by grants awarded to J.M.C from the National Sanitarium Association, the Natural Sciences and Engineering Research Council of Canada (RGPIN-2016-05730), and Canadian Institutes of Health Research (PPE-192116). S.Z. is the recipient of a Devolved Graduate Scholarship from the Vaccinology and Immunotherapeutics Program, School of Public Health, University of Saskatchewan. VIDO receives operational funding from the Canadian Foundation for Innovation through the Major Science Initiatives Program, and the Government of Saskatchewan through Innovation Saskatchewan, and the Ministry of Agriculture. Antibodies against *M. tb* EsxB (NR-13801), GroEL2 (NR-13813), and Antigen-85 complex (NR-13800) were obtained through BEI Resources, NIAID, NIH. This manuscript is published with permission from the Director of VIDO as journal series number 1112.

## AUTHOR AFFILIATIONS

[1]Vaccine and Infectious Disease Organization, Saskatoon, Canada
[2]Vaccinology and Immunotherapeutics Program, School of Public Health, University of Saskatchewan, Saskatoon, Canada

## AUTHOR ORCIDs

Jeffrey M. Chen ⓘ http://orcid.org/0000-0001-8431-3802

## FUNDING

| Funder | Grant(s) | Author(s) |
| --- | --- | --- |
| National Sanitarium Association | | Jeffrey M. Chen |
| Natural Sciences and Engineering Research Council of Canada | RGPIN-2016-05730 | Jeffrey M. Chen |
| Canadian Institutes of Health Research | PPE-192116 | Jeffrey M. Chen |

## AUTHOR CONTRIBUTIONS

Slim Zriba, Conceptualization, Formal analysis, Investigation, Methodology, Writing – review and editing | Ze Long Lim, Formal analysis, Investigation, Writing – review and editing | Marlene Snider, Formal analysis, Investigation | Nirajan Niroula, Formal analysis, Investigation, Writing – review and editing | Marie Hardouin, Investigation | Jeffrey M. Chen, Conceptualization, Formal analysis, Funding acquisition, Investigation, Methodology, Project administration, Resources, Supervision, Writing – original draft, Writing – review and editing

## ETHICS APPROVAL

The mouse trial described in this study was conducted in our biosafety level 3 laboratories at VIDO in accordance with guidelines of the Canadian Council on Animal Care and approved by the Animal Research Ethics Board at the University of Saskatchewan (Animal Use Protocol # 20190059).

## ADDITIONAL FILES

The following material is available online.

### Supplemental Material

**Fig. S1 and S2 (Spectrum01131-25-s0001.pdf).** Fig. S1: Immunoblot analysis of recombinant *M. smegmatis* strains. Immunoblots of (A) 15 µg/well of total CF proteins

and (B) 7.5 µg/well of total CL proteins from *M. smegmatis*::pYUB412 + pMD31, *M. smegmatis*::pYUB412 + pMDespACD, *M. smegmatis*::2F9 + pMD31 and MSX-1 after 10-, 20- and 30-hours of growth in modified 7H9 media.Fig. S2. Immunoblot analysis of MSX-1 cultured in the presence and absence of kanamycin selection pressure. Immunoblots of 10 µg/well of total CL proteins from MSX-1 grown with and without kanamycin selection pressure for 2-, 4-, 6- and 8-days in modified 7H9 media.

## Open Peer Review

**PEER REVIEW HISTORY (review-history.pdf).** An accounting of the reviewer comments and feedback.

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
