## [Reviewer comments · Microbiology Spectrum]

Microbiology Spectrum

Assembly of the *Mycobacterium tuberculosis* type-7 ESX-1 secretion system in *Mycobacterium smegmatis* identifies a new transcriptional activator of *esx-1* genes and a novel TB vaccine

Slim Zriba, Ze Lim, Marlene Snider, Nirajan Niroula, Marie Hardouin, and Jeffrey Chen

Corresponding Author(s): Jeffrey Chen, Vaccine and Infectious Disease Organization International Vaccine Centre

Review Timeline:

Submission Date:	April 15, 2025
Editorial Decision:	May 17, 2025
Revision Received:	July 16, 2025
Accepted:	August 4, 2025

Editor: Olivier Neyrolles

Reviewer(s): Disclosure of reviewer identity is with reference to reviewer comments included in decision letter(s). The following individuals involved in review of your submission have agreed to reveal their identity: Giovanni Delogu (Reviewer #2)

Transaction Report:

DOI: <https://doi.org/10.1128/spectrum.01131-25>

Re: Spectrum01131-25 (**Assembly of the *Mycobacterium tuberculosis* type-7 ESX-1 secretion system in *Mycobacterium smegmatis* identifies a new transcriptional activator of *esx-1* genes and a novel TB vaccine**)

Dear Dr. Jeffrey Matthew Chen:

Thank you for the privilege of reviewing your work. Below you will find my comments, instructions from the Spectrum editorial office, and the reviewer comments.

Your manuscript was reviewed by two experts in the field. While both reviewers are enthusiastic about the study and its overall contribution, they recommend revisions primarily aimed at adding details, clarifications, and expanding the discussion. In particular, Referee 1 raises a relevant concern in Point 1: performing an additional Western blot analysis with a higher protein load could provide clearer and more convincing results.

Revision Guidelines

Sincerely,
Olivier Neyrolles
Editor
Microbiology Spectrum

Reviewer #1 (Comments for the Author):

In this study, Zriba and colleagues reconstructed a functional ESX-1 secretory machinery from *Mycobacterium tuberculosis* in the non pathogenic model organism *Mycobacterium smegmatis*.

The recombinant ESX-1 proficient *M. smegmatis* strain (MSX-1) revealed an intriguing attenuated profile in the THP-1 macrophage model, and resulted as protective as BCG in a mouse immunization-vaccination model. The characterization of the protein profile of *M. smegmatis* strains expressing different variants the *esx-1*-locus and adjacent genes identified Rv3860 as potentially involved in the transcriptional activation of *pe35-ppe68-esxA-esxB* genes.

Overall, all conclusions are supported by the data obtained. However, the manuscript would benefit from a more exhaustive discussion of the obtained results, more comprehensive of observation reported in already published studies.

My specific comments and suggestions are listed below

* Secretion assays, Figure 1. Western blot assays on CF were performed using 5 microgram/well, while the standard amount is 15-25 microgram/well. The low quality of the signal of recognized antigens in these assays makes the comparison of secretion ESX-1 proteins and Ag85 reference antigen among the different recombinant *M. smegmatis* strains quite difficult. Performing the assay with higher amount of CF would improve the quality of the signal, thus corroborating the data interpretation.

* Intracellular survival assays. The expression of the *Mtb espACD* and/or *esx-1* loci does not confer to the corresponding recombinant *M. smegmatis* strains an enhanced survival/growth in infected macrophages as compared to the wild-type strain. How the Authors verified that the replicative plasmid expressing the *espACD* locus is not lost by intracellular *M. smegmatis* strains?

Previous studies reported that the expression of ESX-1 antigens resulted in increased virulence of recombinant *M. smegmatis* both in cellular and mouse models (DOI: 10.1080/22221751.2020.1861913). Moreover, the phagosomal escape ability is mainly dependent on a functional ESX-1 system, with the contribution of phthiocerol dimycocerosates. Recombinant BCG::ESX-1 strains deficient in phthiocerol dimycocerosates are still able to induce phagosomal rupture, although less efficiently than the BCG::ESX-1 exporting phthiocerol dimycocerosates (DOI: 10.1111/cmi.12726). I recommend the Authors to discuss the intracellular survival data also in light of these observations.

*L382-384. What adverse effects the Authors are referring to? How many mice were sacrificed for immunological assays and protection assays?

L387. For clarity, please specify the CFU reduction observed in MSX-1-vaccinated mice as compared to others groups in terms of log-reduction.

*Figure 5. Please report the statistical difference between the CFU numbers in non-vaccinated mice as compared to MSX-1- and BCG-vaccinated mice.

*L42-43; L53-54; L492-493. The PPD skin test is recommended as tuberculosis diagnostic test in selected specific circumstances. Please consider the other WHO/CDC recommended test for TB diagnosis and/or sensitization to Tb antigens when discussing the potential implication of the observed PPD reactions and INF-g responses after an eventual vaccination with the *M. smegmatis* MSX-1 strain.

Reviewer #2 (Comments for the Author):

Comments:

Regarding the experiments to assess the vaccine efficacy, while it is indicated that a more complete characterization is ongoing and will be presented in a separate report, I think that it would be important to discuss not only the DTH response but the antigen specific response. Did the authors investigate the immune responses of the vaccine under study, including the two *M. smegmatis* strains tested, for their ability to stimulate T cell responses against ESX-1 antigens as for instance ESXA&B?

Re: Spectrum01131-25 (Assembly of the *Mycobacterium tuberculosis* type-7 ESX-1 secretion system in *Mycobacterium smegmatis* identifies a new transcriptional activator of *esx-1* genes and a novel TB vaccine)

Response to reviewer 1.

Point 1. Secretion assays, Figure 1. Western blot assays on CF were performed using 5 microgram/well, while the standard amount is 15-25 microgram/well. The low quality of the signal of recognized antigens in these assays makes the comparison of secretion ESX-1 proteins and Ag85 reference antigen among the different recombinant *M. smegmatis* strains quite difficult. Performing the assay with higher amount of CF would improve the quality of the signal, thus corroborating the data interpretation.

We appreciate this reviewer's comments on this point. We opted to run 5 µg/well of CF and 2.5 µg/well of CL proteins as we have done in previous published studies (DOI: 10.1021/acsinfecdis.0c00741) to mitigate signal saturation during detection with the LiCor system and to be able to better detect finer differences in the levels of secreted ESX-1 proteins in the CFs of the different strains. Nevertheless, we have acted on this reviewer's suggestion and repeated the experiment with fresh cultures and examined levels of the proteins of interest in total CF and CL proteins at higher amounts as recommended. The new data which remains consistent with the original data is discussed and presented as a supplemental figure for the revised manuscript.

Point 2. Intracellular survival assays. The expression of the *Mtb espACD* and/or *esx-1* loci does not confer to the corresponding recombinant *M. smegmatis* strains an enhanced survival/growth in infected macrophages as compared to the wild-type strain. How the Authors verified that the replicative plasmid expressing the *espACD* locus is not lost by intracellular *M. smegmatis* strains?

This is a valid point and while we acknowledge that pMDespACD (episomal) may be lost in the absence of kanamycin selection pressure, we do not believe this to be the case with MSX-1 at least for the duration of the macrophage infection experiments based on 2 lines of evidence – 1) we recovered and enumerated CFUs of intracellular MSX-1 from THP-1 cells on 7H10 agar supplemented with hygromycin and kanamycin, 2) We cultured MSX-1 in growth media lacking kanamycin and found EspD expression to be comparable to what is seen in MSX-1 grown in the presence of kanamycin for up to 8 days – this data is discussed and presented as a supplemental figure in the revised manuscript.

Point 3. Previous studies reported that the expression of ESX-1 antigens resulted in increased virulence of recombinant *M. smegmatis* both in cellular and mouse models (DOI: 10.1080/22221751.2020.1861913).

The recombinant *M. smegmatis* strain characterized by Guo et al. (DOI: 10.1080/22221751.2020.1861913) is markedly different from MSX-1. Their strain expresses *M. tb* EspC alone under the control of an IPTG-inducible promoter in the pPscreen shuttle vector. MSX-1 on the other hand expresses EspA, EspC and EspD under the control of the native *espA* promoter in pMDespACD along with all of the other ESX-1 proteins encoded in the cosmid 2F9 to produce the entire *M. tb* ESX-1 machinery. Furthermore, Guo et al. infected RAW264.7 cells with their strain at a much higher multiplicity of infection (MOI =10) whereas we infected THP-1 cells with MSX-1 at an MOI = 0.25 to measure intracellular replication. Guo et al. also injected 10⁷ and 10⁸ CFUs of their *M. smegmatis* strain intravenously (via the tail vein), a route of infection that is known to be lethal in mice even with wild-type *M. smegmatis* (DOI: 10.1038/nm.2420), to assess replication in lungs and spleens, and survival respectively. In contrast, we injected mice with 10⁶ CFUs of our *M. smegmatis* strains subcutaneously. These differences in the nature of the genetic

modification, macrophage models of infection and inoculation dose taken together likely account for the contrasting observations.

Point 4. Moreover, the phagosomal escape ability is mainly dependent on a functional ESX-1 system, with the contribution of phthiocerol dimycocerosates. Recombinant BCG::ESX-1 strains deficient in phthiocerol dimycocerosates are still able to induce phagosomal rupture, although less efficiently than the BCG::ESX-1 exporting phthiocerol dimycocerosates (DOI: 10.1111/cmi.12726).

We had discovered previously that BCG Japan and Moreau are deficient in PDIMs and PGL (DOI: 10.1016/j.vaccine.2007.09.041). When BCG Japan and Moreau were transformed with the *M. tb* *esx-1* locus as described in the study referenced by this reviewer (DOI: 10.1111/cmi.12726) both BCG Japan::ESX-1 and BCG Moreau::ESX-1 in contrast to PDIM-proficient BCG Pasteur::ESX-1, remain unable to induce phagosomal rupture in THP-1 cells (see Fig. 5 C-D in DOI: 10.1111/cmi.12726). This phenotype is actually consistent with our contention that despite expressing a functional *M. tb* ESX-1 system, MSX-1 is non-cytopathic likely because it does not make PDIMs which is now known to synergize with the T7SS to mediate full virulence.

Point 5. L382-384. What adverse effects the Authors are referring to? How many mice were sacrificed for immunological assays and protection assays?

Mice were monitored daily by our veterinary team in infection trials for appearance and behaviour and scored according to a scale (0- Normal; 1- Lack of grooming; 2- Rough coat, hunched but still mobile; 3- Rough coat, hunched and no longer mobile). They were also weighed at regular intervals to determine if they are losing weight and not thriving despite provision of feed and water ad libitum. These criteria helped them decide if animals had reached humane endpoint. In this study, all mice regardless of the vaccine treatment group scored “Normal” and gained weight throughout the entirety of the trial. We have now added this information in the revised manuscript. Each vaccination group (including the mock group) consisted of 8 mice – 2 mice were sacrificed for the splenocyte assay with the remaining 6 mice challenged with *M. tb* to assess for vaccine-mediated protection. Of these 6, organs from 4 were assessed for bacterial burden by CFU plating and 2 for H/E staining. This breakdown is described in the material and methods section.

Point 6. L387. For clarity, please specify the CFU reduction observed in MSX-1-vaccinated mice as compared to other groups in terms of log-reduction.

Done. We have now clarified this in the results section of the revised manuscript.

Point 7. Figure 5. Please report the statistical difference between the CFU numbers in non-vaccinated mice as compared to MSX-1- and BCG-vaccinated mice.

Done. This is now stated in the results section of the revised manuscript.

Point 8. L42-43; L53-54; L492-493. The PPD skin test is recommended as tuberculosis diagnostic test in selected specific circumstances. Please consider the other WHO/CDC recommended test for TB diagnosis and/or sensitization to Tb antigens when discussing the potential implication of the observed PPD reactions and INF-g responses after an eventual vaccination with the *M. smegmatis* MSX-1 strain.

We assume this reviewer is referring to the QuantiFERON test which uses EsxA (ESAT-6) and EsxB (CFP-10) as recall antigens and can differentiate individuals vaccinated with BCG from those with *M. tb* infection. Although in this study we did not use EsxA as recall antigen in mouse splenocyte assays, we did do so in a follow-up study. Consistent with the lack of sensitization to tuberculin (which contains EsxA and EsxB amongst other major MTBC secreted antigens) seen in this study, mice vaccinated with MSX-1 did not become sensitized to an EsxA peptide containing immunodominant epitopes in splenocyte assays (please refer to unpublished data in

response to reviewer 2). This suggests that MSX-1 vaccinees will likely also not test positive with the QuantiFERON test. Changes reflecting this have been made in the revised manuscript.

Response to reviewer 2.

Regarding the experiments to assess the vaccine efficacy, while it is indicated that a more complete characterization is ongoing and will be presented in a separate report, I think that it would be important to discuss not only the DTH response but the antigen specific response. Did the authors investigate the immune responses of the vaccine under study, including the two *M. smegmatis* strains tested, for their ability to stimulate T cell responses against ESX-1 antigens as for instance ESXA&B?

In this study we did not specifically use EsxA or EsxB as recall antigens in mouse splenocyte assays. In a follow-up study however (manuscript in prep.), we assessed and compared the DTH response after vaccination with either BCG::pYUB412 (BCG with empty vector control), BCG::ESX-1 (BCG complemented with 2F9), *M. smegmatis*::pYUB412+pMD31, MSX-1 and a modified MSX-1 called MSX-1 V2, to PPD and EsxA peptide containing immunodominant epitopes. Consistent with the lack of sensitization to PPD (which contains EsxA and EsxB as well as other major MTBC secreted antigens) seen in this study, splenocytes from mice vaccinated with MSX-1, MSX-1 V2 and *M. smegmatis*::pYUB412+pMD31 were not sensitized to either PPD or to the EsxA peptide. In contrast but as expected, splenocytes from mice vaccinated with BCG::pYUB412 and BCG::ESX-1 became sensitized to PPD, but splenocytes only from mice vaccinated with BCG::ESX-1 became sensitized to the EsxA peptide (see unpublished figure below).

Splenocyte stimulation assay. Splenocytes from mock vaccinated mice and mice vaccinated with BCG::pYUB412 (wild-type), BCG::ESX-1, *M. smegmatis*::pYUB412+pMD31 (wild-type), MSX-1 and MSX-1 V2 were incubated with media only, media containing ConA (1 µg/mL), EsxA peptide (1 µg/mL) and PPD (5 µg/mL). IFN γ produced and released into the supernatant was quantified by ELISA.

Re: Spectrum01131-25R1 (**Assembly of the *Mycobacterium tuberculosis* type-7 ESX-1 secretion system in *Mycobacterium smegmatis* identifies a new transcriptional activator of *esx-1* genes and a novel TB vaccine**)

Dear Dr. Jeffrey Matthew Chen:

Your manuscript has been accepted, and I am forwarding it to the ASM production staff for publication. Your paper will first be checked to make sure all elements meet the technical requirements. ASM staff will contact you if anything needs to be revised before copyediting and production can begin. Otherwise, you will be notified when your proofs are ready to be viewed.

Sincerely,
Olivier Neyrolles
Editor
Microbiology Spectrum

Reviewer #1 (Comments for the Author):

In the revised version of the manuscript, the Authors the Authors answered all points raised by the reviewers.